# Tissue specific LRRK2 interactomes reveal a distinct striatal functional unit

Yibo Zhao[1], Nikoleta Vavouraki[2¤], Ruth C. Lovering[3], Valentina Escott-Price[4], Kirsten Harvey[1], Patrick A. Lewis[2,5,6], Claudia Manzoni[1] *

**1** University College London, School of Pharmacy, London, United Kingdom, **2** University of Reading, School of Pharmacy, Reading, United Kingdom, **3** University College London, Institute for Cardiovascular Science, London, United Kingdom, **4** University of Cardiff, School of Medicine, Division of Psychological Medicine and Clinical Neurosciences, Cardiff, United Kingdom, **5** Royal Veterinary College, London, United Kingdom, **6** UCL Queen Square Institute of Neurology, London, United Kingdom

¤ Current address: Oxford-GSK Institute of Molecular and Computational Medicine Nuffield Department of Medicine, University of Oxford, Oxford, United Kingdom
* c.manzoni@ucl.ac.uk

**Data Availability Statement:** All relevant data are within the manuscript and its Supporting Information files.

## Abstract

Mutations in *LRRK2* are the most common genetic cause of Parkinson's disease. Despite substantial research efforts, the physiological and pathological role of this multidomain protein remains poorly defined. In this study, we used a systematic approach to construct the general protein-protein interactome around LRRK2, which was then evaluated taking into consideration the differential expression patterns and the co-expression behaviours of the LRRK2 interactors in 15 different healthy tissue types. The LRRK2 interactors exhibited distinct expression features in the brain as compared to the peripheral tissues analysed. Moreover, a high degree of similarity was found for the LRRK2 interactors in *putamen*, *caudate* and *nucleus accumbens*, thus defining a potential LRRK2 functional cluster within the *striatum*. The general LRRK2 interactome paired with the expression profiles of its members constitutes a powerful tool to generate tissue-specific LRRK2 interactomes. We exemplified the generation of the tissue-specific LRRK2 interactomes and explored the functions highlighted by the "core LRRK2 interactors" in the *striatum* in comparison with the *cerebellum*. Finally, we illustrated how the LRRK2 general interactome reported in this manuscript paired with the expression profiles can be used to trace the relationship between LRRK2 and specific interactors of interest, here focusing on the LRRK2 interactors belonging to the Rab protein family.

## Author summary

Parkinson's disease is characterized by the progressive degeneration of the brain (neurodegeneration) involving the brain regions responsible for controlling fine movements. The exact mechanism responsible for Parkinson's neurodegeneration is still unclear, however different mutations in multiple genes have been associated with the disease. Mutations in the *LRRK2* gene constitute a genetic risk factor for both familial and sporadic Parkinson's disease, suggesting that the LRRK2 protein may play an important role in

**Funding:** YZ is supported by a studentship from the UCL School of Pharmacy. PAL and CM received funding from the Biomarkers Across Neurodegenerative Diseases Grant Program 2019, BAND3 (Michael J. Fox Foundation, Alzheimer's Association, Alzheimer's Research UK, and the Weston Brain Institute [grant number 18063]), this funding partially covered CM and PAL salaries. NV PhD and salary were supported by a grant awarded to PAL from the Engineering and Physical Sciences Research Council (studentship EP/M508123/1) and by the Dolby Family Fund. VEP receives funding from the UK DRI funded by the UK Medical Research Council (UKDRI-3003) and the Joint Programming for Neurodegeneration (JPND) - (MRC: MR/T04604X/1); these fundings partially cover VEP salary. RCL is funded via the National Institute for Health Research University College London Hospitals Biomedical Research Centre. The funders had no role in study design, data collection and analysis, decision to publish, or preparation of the manuscript.

**Competing interests:** NV received a scholarship award from Keystone Symposia and an award from Alzheimer's Research UK. RCL received funding from Alzheimer's Research UK. KH received funding from Parkinson's UK, PharmAlliance, Wellcome Trust-NIH, MRC (M013502 and M00676X). PAL acted as a paid consultant to Merck Sharpe and Dohm; received funding from ASAP, MJFF, MRC and Parkinson's UK; is member of BBSRC committee D, chair of the MS Society catalyst funding panel, member of the MS Society Research Strategy Committeee, member of Parkinson's UK pool of experts, member of the NC3Rs studentship panel. CM received honoraria for lecturing at the University of Padova and reimbursements for lecturing at the Azienda ULSS2 Marca Trevigiana; travel grants from Alzheimer's Research UK, the University of Reading, Guarantors of Brain and the Biochemical Society; funding from EMBO and MJFF.

Parkinson's. It has been suggested LRRK2 to be a "hub protein", meaning a protein involved in the orchestration of multiple functions within the cell; functions that are probably activated with regional and temporal specificity. Here, we constructed a computational model centred on LRRK2 by collecting information related to the hundreds of proteins that have been found able to interact with LRRK2 (LRRK2 interactome). This model can be used to simulate LRRK2 functions in different tissues, additionally it can be used to trace specific LRRK2 interactors and follow their behaviour across different tissues. For example, we showed a dichotomy for the LRRK2 interactome expression between the brain and the peripheral tissues; while the brain regions forming the striatum (the structure mostly affected in Parkinson's) showed a similar functional profile.

## Introduction

Leucine-rich repeat kinase 2 (LRRK2) is a large (285kDa), multidomain protein. As a member of the ROCO superfamily, LRRK2 contains a Ras-of complex (ROC) GTPase domain, as well as a serine-threonine kinase domain, linked to the ROC domain by a C-terminal-of-ROC (COR) domain of unclear function. This enzymatic core is flanked by 4 protein-protein interaction domains [1]. This complex domain structure makes LRRK2 an interesting enzyme from a biochemical perspective, suggesting it may act as a signalling hub able to orchestrate different cellular functions. Mutations in the *LRRK2* gene are the most common genetic causes of familial Parkinson's disease (PD), accounting for 2–40% of cases depending on the population under analysis [2], with the most common (and most intensively studied) mutation being a pathogenic G2019S amino acid change located in the kinase domain [3]. This mutation has been associated with increased LRRK2 kinase activity, and has been implicated in pathophysiological changes including lysosomal dysfunction, α-synuclein and tau aggregations and dysregulation of neuroinflammation [4]. However, although many years of study have generated a vast array of results, the role(s) of LRRK2 in health and disease still remain(s) elusive.

Apart from PD, multiple lines of evidence have associated LRRK2 with a number of peripheral diseases induced by excessive inflammatory response. *LRRK2* has been identified as a major susceptibility gene for Crohn's disease (CD) [5–9]. A newly identified LRRK2-N2081D mutation, which is located in the kinase domain, is potentially associated with increased risk for both PD and CD [10]. Additionally, LRRK2 mutations were reported to aggravate the type-1 reaction in leprosy, and the innate immune response against *Mycobacterium tuberculosis* [11,12]. These findings indicate a potentially important role of LRRK2 at the interface between the peripheral and the central nervous system (CNS) immunity. Mutations in LRRK2 have also been linked to an increased risk of cancer [13]. Clinical studies found that PD patients carrying the LRRK2-G2019S mutation present higher risk of non-skin cancer, hormone-related cancers and breast cancer compared to non-carriers [14,15].

Although the mechanism underlying the link between LRRK2 and CD, leprosy and cancer is still unclear, this substantial association with peripheral diseases suggests a probably equally important role of LRRK2 throughout the body, both within and beyond the CNS.

Protein-protein interactions (PPIs) are fundamental for the maintenance of cellular homeostasis, with their alterations (due to mutations or post translational modification) potentially leading to diseases [16,17]. Systems biology deals with the complexity of PPIs applying approaches based on a holistic rather than on a one-to-one perspective. In particular, the core assumption is that proteins interacting with each other constitute a functional unit and thereby are more likely to cooperate in the same cellular pathway(s). In this perspective, the analysis of

the protein network built around one or more "seed proteins" of interest allows to gain insights into the biological processes sustained by them, as a community. In this study, LRRK2 was designated as the "seed protein" at the centre of the network analysis, and its interactors were derived from curated peer-reviewed literature. The obtained LRRK2 interactome was then investigated with the goal of describing the cellular functions regulated by LRRK2 together with its interactors under physiological conditions. This strategy had already been employed by 2 comprehensive LRRK2-focused PPI studies which were independently published in 2015 [18,19]. These studies, however, did not investigate the tissue specificity of the LRRK2 interactome.

Multiple lines of evidence have shown that the expression levels of LRRK2 differs greatly in different tissues and cell types, with one possible implication that this variability in expression levels may reflect underlying divergence in the cellular functions of LRRK2 [20–27]. From a cellular perspective, LRRK2 has been associated with many different functions, ranging from modulation of autophagy to control of vesicles dynamics, and from regulation of signalling pathways to response to stressors [13,28–33]. One strategy to make sense of this plethora of functions, is to suggest that LRRK2 might potentially be implicated in different processes in different tissues, thus reflecting a tissue specific (and potentially cell-specific) functional profile, which might be a consequence of the existence of tissue specific LRRK2 protein complexes [34].

Therefore, here we hypothesized that the interactions between LRRK2 and its partners might be tissue specific. However, since most of PPI data currently available are principally derived from *in-vitro* experiments in cellular models, isolated proteins or from high through-put screening, they lack in tissue specificity. With this work we propose computational approaches to differentiate the general LRRK2 interactome into tissue specific LRRK2 interactomes considering the transcriptomic features and the functional patterns of LRRK2 and its interactors in healthy human tissues. These results provide a tool to model tissue specificity for PPIs and a valuable window onto the role of LRRK2 in health and disease with important implications for the development of safe LRRK2-targeted therapeutic approaches.

## Methods

### Protein-protein interaction (PPI) download

PINOT (v1.1) [35] (http://www.reading.ac.uk/bioinf/PINOT/PINOT_form.html), HIPPIE (v2.2) (http://cbdm-01.zdv.uni-mainz.de/~mschaefer/hippie/index.php) [36]and MIST (v5.0) (https://fgrtools.hms.harvard.edu/MIST/) [37] were queried to download "homo sapiens" PPIs for LRRK2 (UniProt ID: Q5S007, 21 October 2020). To access the broadest possible set of PPI data, "Lenient" filter level was applied in PINOT; while all filters were removed in HIPPIE and MIST.

PPIs were quality controlled via an in-house pipeline described in **Fig 1**: 1) protein IDs of data from different repositories were converted into the same identifier system (HUGO Gene Nomenclature Committee (HGNC) gene symbols); 2) the "interaction detection methods" in HIPPIE and MIST were reassigned referring to the PINOT Method Grouping Dictionary (Lenient version, **Table A in S1 Table**). The PINOT Method Grouping Dictionary clusters similar detection methods annotated in PSI-MI ontology (e.g. "Two hybrid fragment pooling approach MI:0399" and "Two hybrid bait and prey pooling approach MI:1113" are allocated in the same category: "Two Hybrid"); 3) PPIs extracted from the 3 databases were merged after removing duplicates; and 4) LRRK2 interactors were then scored (Final Score, FS) by adding the number of detection methods (Method Score, MS) and the number of reporting publications (Publication Score, PS). The LRRK2 interactome was generated from interactors with

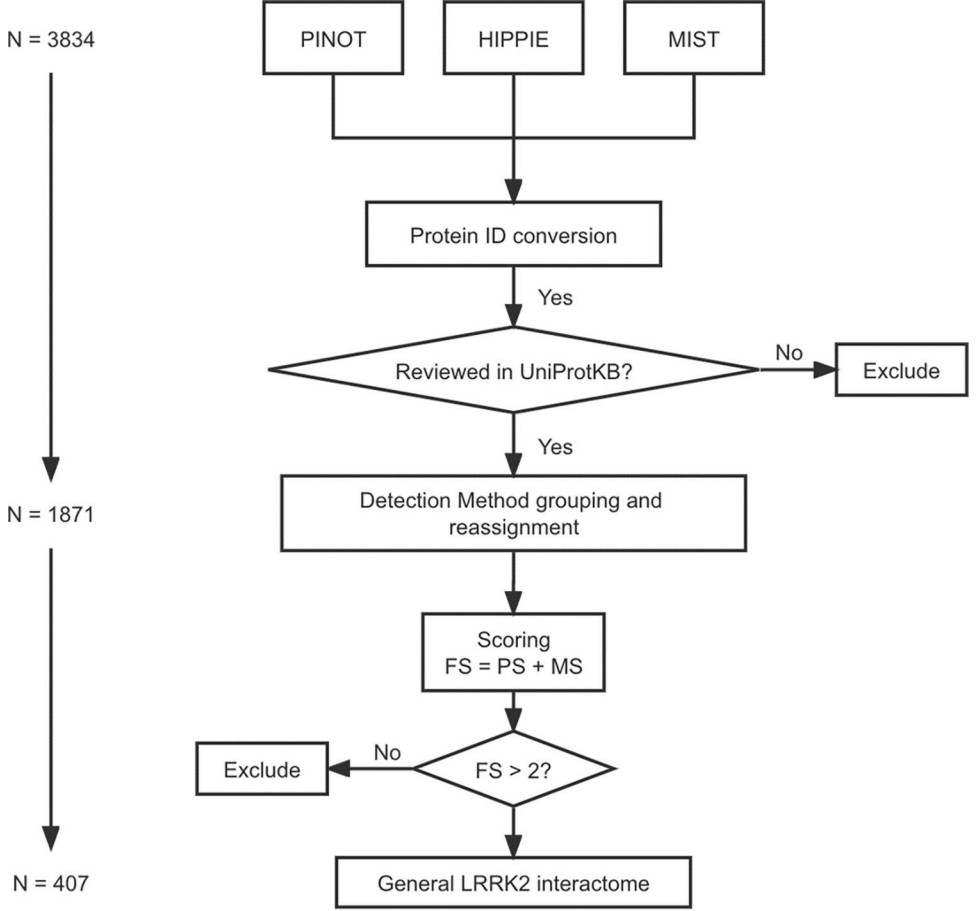

**Fig 1. PPI quality control pipeline for LRRK2 interactome construction.** Human PPI data downloaded from PINOT, HIPPIE and MIST databases were merged after ID conversion using HGNC gene symbols. Merged data underwent interaction detection method reassignment using an in-house dictionary. Publication score (PS) was defined as the number of papers in which a PPI was reported, while method score (MS) was defined as the number of different methods by which a PPI was detected. Final score (FS) was calculated as PS + MS. PPIs with FS $\leq$ 2 were excluded from further analysis.

FS > 2. Interactors with lower FS ($\leq$ 2) were removed from further analysis because of their poor reliability (either were not replicated in multiple experiments or with missing publication identifier or with missing record of detection method). Of note, interactors marked as "Unreviewed" in UniProtKB were removed as well. PPIs that passed quality control thereby constitute what we defined as the general LRRK2 interactome (LRRK2$_{int}$). Family classification of QC-ed LRRK2 interactors was extracted (on 19 December 2020) from UniProt via the R package: UniProt.ws [38].

## Functional enrichment

The general LRRK2$_{int}$ was analysed by Functional Enrichment Analysis using the tool g:GOSt on 8 November 2022 (g:Profiler (https://biit.cs.ut.ee/gprofiler/gost) [39]. The parameters were set as follows: organism—Homo sapiens (Human); data source—GO biological process (GO-BPs) only; statistical domain scope (i.e. background)–annotated genes only; statistical method—Fisher's one-tailed test; significance threshold–Bonferroni correction (threshold = 0.05). No hierarchical filtering was included. To increase the sensitivity of analysis, a cut-

off of ≤ 1500 was set for the "term size" (in the annotated human genome) of enriched GO terms. Finally, GO-BPs whose enrichment gene set did not include LRRK2 were discarded, thereby keeping only the GO-BPs to which LRRK2 directly contributed. The remaining GO-BPs were grouped according to their semantic similarity (hereby referred as "GO-BP groups") using an in-house dictionary followed by manual scrutiny. LRRK2 interactors contributing to the enrichment of each GO-BP group were extracted (hereby referred as "functional groups"). The composition (in terms of LRRK2 interactors) of different functional groups was compared using Multiple Correspondence Analysis (FactoMineR and Factoextra R packages).

The tissue specific LRRK2$_{ints}$ and Co-ex/DEA_Cluster$_{LRRK2}$ were analysed by Functional Enrichment Analysis using the tool g:GOSt on 21 Oct 2022 using the same procedure as above. In this case we have reduced the background noise to a minimum by removing electronic GO annotations and by filtering for term size ≤ 80.

## RNA-Seq data download and quality control

RNA-Seq data (read counts) were downloaded for 11 brain regions (amygdala, anterior cingulate cortex, caudate (basal ganglia), cerebellum/cerebellar hemisphere, cortex/frontal cortex (BA9), hippocampus, hypothalamus, nuclear accumbens (basal ganglia), putamen (basal ganglia), spinal cord (cervical c-1) and substantia nigra (basal ganglia)), 3 peripheral tissues (lung, liver, kidney) and whole blood from the Genotype-Tissue Expression (GTEx, https://www.gtexportal.org/home/) Analysis Release V8 (dbGaP Accession phs000424.v8.p2) on 19 Aug 2021 (https://storage.googleapis.com/gtex_analysis_v8/rna_seq_data/GTEx_Analysis_2017-06-05_v8_RNASeQCv1.1.9_gene_reads.gct.gz). Of note, in GTEx, "cerebellum/cerebellar hemisphere" and "cortex/frontal cortex (BA9)" are duplicated pairs (https://www.gtexportal.org/home/faq - brainCortexAndCerebellum). Therefore, we kept "cerebellum" and "cortex" in our analysis because they contain more samples compared to their duplicates (209 vs. 175; 205 vs. 175). For kidney, we only kept data for cortex, while data for medulla were discarded because it contains too few samples (n = 4).

RNA-seq data was then quality controlled following an in-house pipeline: for each tissue, 1) low count genes (with read counts = 0 in more than 5% of all samples) were discarded; 2) samples with mapping rates < 80% were discarded (GTEx's Sample Attribute file: https://storage.googleapis.com/gtex_analysis_v8/annotations/GTEx_Analysis_v8_Annotations_SampleAttributesDS.txt); 3) expression-profile similarity among remaining samples in a given tissue was examined via pairwise Pearson's correlation test on the read counts; 4) samples were clustered based on the correlation coefficients calculated in 3). Samples that lay outside the main cluster(s) in the hierarchical dendrogram were recognised as "outliers" and were thereby discarded from further analysis. Read counts for LRRK2 interactors were extracted using HGNC gene symbols and Ensembl Gene IDs. LRRK2 interactors with available expression data for all tissues were kept for further analysis.

## Differential expression analysis

Pair-wise Differential Expression Analysis (DEA) was performed to compare the expression values of LRRK2 interactors across different tissues via the R package DESeq2 [40]. In DESeq2, p-values generated during each round of DEA between every pair of tissues were automatically adjusted for multiple testing correction. These p-values were further corrected using Bonferroni's method to reduce the Type II error generated from pair-wise DEA among all tissues of analysis. This resulted in a matrix containing DEA p-values for each possible pair of tissues, for all LRRK2 interactors. For each interactor, tissues were ranked according to the

DEA results via the following method: for interactor $I$, if its expression level in Tissue $A$ is significantly higher than in Tissue B (Bonferroni corrected p-value $< 0.05$), then $Score_{I,A}^{Ex} = Score_{I,A}^{Ex} + 1$, while $Score_{I,B}^{Ex}$ remains unchanged, and vice versa. If the comparison between Tissue $A$ and Tissue $B$ is non-significant (Bonferroni corrected p-value $\geq 0.05$), both $Score_{I,A}^{Ex}$ and $Score_{I,B}^{Ex}$ remain unchanged. According to this classification, for interactor $I$, $Score_{I,A}^{Ex}$ was directly indicative of the expression level of interactor $I$ in Tissue $A$ in comparison with the other tissues: the higher the $Score_{I,A}^{Ex}$, the higher the expression of interactor $I$ in Tissue $A$. Some interactors presented uniquely and significantly high expression in certain tissues ($Score_{I,A}^{Ex} \geq 12$), meaning that the expression level of Interactor $I$ in Tissue $A$ was higher than 85.7% (12/14) tissues.

Finally, a heatmap (Heatmap_DEA) was generated based on the mean expression values of LRRK2 interactors in different tissues calculated from the normalised read counts matrix derived from DEA using the R package gplots (DESeq2 transforms the read counts via internal normalisation where geometric mean is calculated for each gene across all samples) [40]. Two dendrograms were derived from Heatmap_DEA: 1) hierarchical clustering of tissues based on the similarity of LRRK2$_{int}$'s expression distribution (Den_DEA1); 2) hierarchical clustering of LRRK2 interactors based on the similarity of their expression patterns across different tissues (Den_DEA2). By cutting these 2 dendrograms, we identified 1) clusters of tissues in which the LRRK2$_{int}$ presented similar expression behaviours; 2) clusters of interactors that exhibited similar expression patterns with LRRK2 across different tissues (DEA_Cluster$_{LRRK2}$).

## Co-expression analysis

Read counts data derived from GTEx was used to calculate Pearson's correlation between LRRK2 and its interactors in each single tissue. Multiple testing correction was performed using Bonferroni's method. The co-expression patterns of LRRK2 with its interactors in each tissue were evaluated by comparison to the corresponding reference co-expression coefficient distribution in the same tissue, which was generated (for each tissue) from co-expression analysis between LRRK2 and 1000 sets of randomly picked genes (to match the size of the general LRRK2$_{int}$) in GTEx.

The distribution of the co-expression coefficients for the LRRK2 interactors was compared across different tissues via the following steps: 1) One-way ANOVA followed by Tukey's test was performed to compare the coefficients across tissues; 2) if the co-expression coefficients were significantly higher in Tissue $A$ than in Tissue $B$ (adjusted p-value $< 0.05$), then $Score_{A}^{coex} = Score_{A}^{coex} + 1$, while $Score_{B}^{coex}$ remained unchanged. If the comparison between Tissue $A$ and Tissue $B$ was non-significant (adjusted p-value $\geq 0.05$), both $Score_{A}^{coex}$ and $Score_{B}^{coex}$ remained unchanged. According to this method, for interactor $I$, the $Score_{A}^{coex}$ in was directly indicative of the co-expression level of interactor $I$ with LRRK2 in Tissue $A$.

Finally, a heatmap (Heatmap_Co-ex) was generated based on the matrix of co-expression coefficients between interactors and LRRK2 in different tissues using the R package gplots. Two dendrograms were derived from Heatmap_Co-ex: 1) hierarchical clustering of tissues based on the similarity of LRRK2-interactor coefficient distribution (Den_Co-ex1); 2) hierarchical clustering of LRRK2 interactors based on the similarity of their co-expression (with LRRK2) behaviours across different tissues (Den_Co-ex2). By cutting the 2 dendrograms, we identified 1) clusters of tissues in which the LRRK2$_{int}$ presented similar co-expression behaviours; 2) clusters of interactors that exhibited similar co-expression patterns with LRRK2 across different tissues. In addition, we compared the co-expression coefficients of the interactor clusters identified in 2) via t-test, thereby identifying the cluster of interactors that presented the highest co-expression with LRRK2 (Co-ex_Cluster$_{LRRK2}$).

## Results

### Construction of the general LRRK2$_{int}$

A total of 1436, 548 and 1850 human LRRK2 interactors were retrieved from PINOT, HIPPIE and MIST respectively (**Fig 1**). To harmonise different protein identifiers, all protein IDs were converted to HGNC gene symbols. One interactor (Entrez ID: 333931) was removed because its record is no longer valid in NCBI Gene. Furthermore, 3 proteins coded by transcriptional read-throughs (RPL17-C18orf32, TPTEP2-CSNK1E and BUB1B-PAK6) were marked as "Unreviewed" entries in UniProtKB, and were thereby discarded from further analysis. After protein ID conversion, the 3 protein sets (3830 annotations in total) were merged into 1 list of 1871 unique interactors (hereby referred as "merged list", **Table B in S1 Table**), suggesting that albeit the differences in data sources and versions, there is a large amount of overlap among PINOT, HIPPIE and MIST in terms of PPIs retrieved for LRRK2. Of note, among the 1871 interactors, 529 (28.3%) were shared by all the 3 tools; 902 (48.2%) were downloaded from 2 of the 3 databases; 440 (23.5%) were present in only 1 database (10 in PINOT, 11 in HIPPIE and 419 in MIST, **Fig A in S1 Fig**).

For each PPI in the merged list, the "interaction detection methods" were extracted and reassigned according to PINOT Method Grouping Dictionary (Lenient version, **Table A in S1 Table**), where similar methods are grouped together. In this way only technically different interaction detection methods were considered as independent evidence of protein interaction, thus adding stringency to the PPI collection pipeline.

The LRRK2 interactors were scored based on the number of publications (Publication Score, PS) and different types of detection methods (Method Score, MS), thereby generating the Final Score (FS = PS + MS). A total of 1463 "low-quality" interactors (1464/1871, 78.2%) with an FS $\leq$ 2 (indicating that these LRRK2 interactions were reported in 1 publication and with 1 method only, therefore never replicated) were identified and removed from further analyses.

A final list containing 407 LRRK2 interactors with FS > 2 was obtained (hereby referred as "the general LRRK2$_{int}$", **Table C in S1 Table**). Among the 407 interactors, 348 (85.6%) were scored FS $\leq$ 5; 41 (10%) were scored between 6 and 8; 18 (4%) were scored FS $\geq$ 9. LRRK2 itself exhibited the highest FS = 50, as many publications confirmed LRRK2 as able to self-interact. Other robust LRRK2 interactors were HSP90AA1 (FS = 19); YWHAQ/14-3-3T (FS = 14), followed by HSPA8, MSN, YWHAZ/14-3-3Z, CDC37, DNM1L, STUB1 and TUBB (**Fig 2A**).

Out of the 407 LRRK2 interactors, 232 (56.9%) were classified into families based on Uni-Prot record: cytoskeleton proteins (n = 51 interactors, 12.5%), ribosomal proteins (n = 41 interactors, 10.1%), protein kinases (n = 35 interactors, 8.6%), GTPases (n = 24 interactors, 5.9%), ATPases (n = 14 interactors, 3.4%), heat shock proteins (n = 12 interactors, 2.9%), and mitochondrial carriers (n = 8, 2.0%). In addition, 12 (2.9%) LRRK2 interactors were classified in ubiquitin-proteasome related protein families; while 27 interactors (6.6%) belonged to gene expression-related families (8 transcription factors/regulators, 5 helicases, 4 splicing factors and 10 DNA-metabolism-related proteins) (**Fig 2B** and **Table D in S1 Table**). Of note, the LRRK2 interactors included 13 Rab GTPases and seven 14-3-3 proteins. The Rab GTPase family and the 14-3-3 family are the two groups of proteins that have been most widely reported as LRRK2 interactors [41,42].

### Functional enrichment analysis of the general LRRK2$_{int}$.

Functional Enrichment Analysis was performed for the general LRRK2$_{int}$. A total of 480 significant GO-BPs (Bonferroni adjusted p-values < 0.05) were returned (**Table E in S1 Table**). A

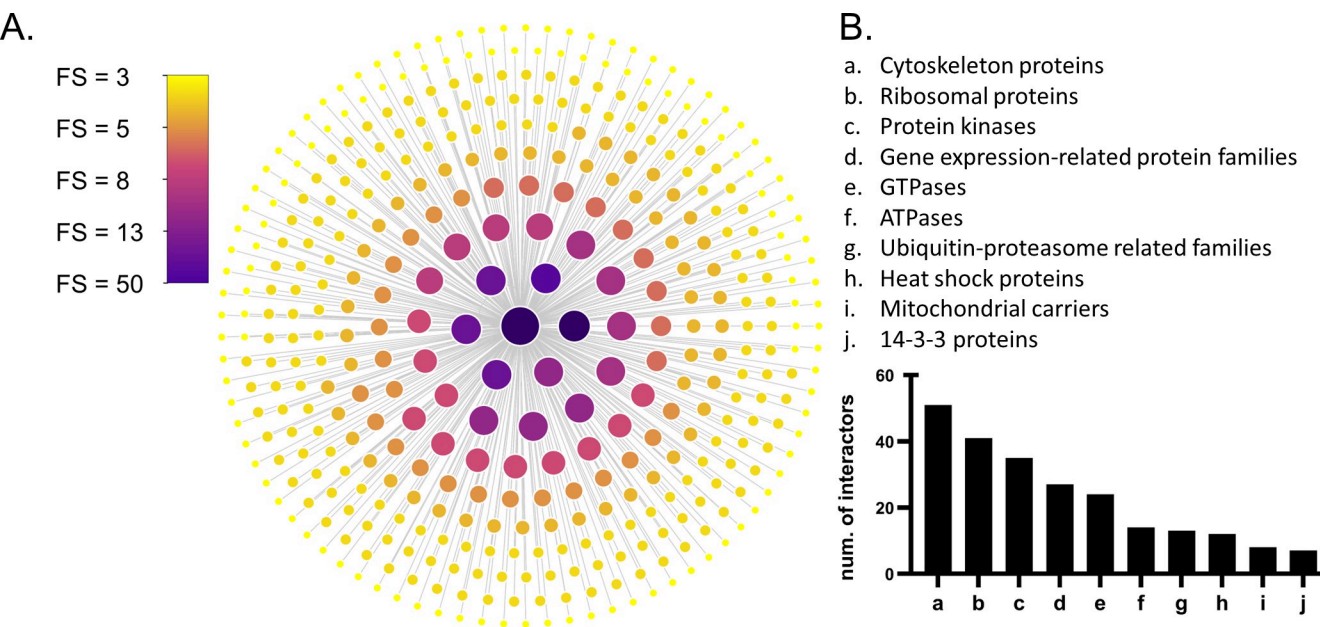

**Fig 2. The general LRRK2 interactome.** A) Nodes (n = 407) represent LRRK2 interactors that were with FS > 2 and were reviewed by the UniprotKB. Node fill colour and node size are weighted on the final score (FS). Larger size and darker colour indicate higher FS. B) Family classification of LRRK2 interactors (of note 176 LRRK2 interactors were not classified into families).

cut-off of "term size" ≤ 1500 was applied to the enriched GO-BPs to remove general terms (126/480, 26.2% of terms removed) and improve sensitivity of analysis. GO-BPs whose enrichment gene set (i.e. intersection) did not contain LRRK2 were also removed leading to the final enrichment list containing 175 GO-BPs (**Table F in S1 Table**). These GO-BPs were clustered into the following 13 functional groups based on their semantic similarities: "autophagy", "cell death", "development, "intracellular organisation", "metabolism", "protein catabolism", "protein localisation", "protein modification", "response to stress", "regulation of enzyme functions", "regulation of gene expression", "signalling", and "transport".

LRRK2 interactors contributing to the enrichment of each single functional group were extracted (**Table G in S1 Table**). A total of 302 (74.2%) LRRK2 interactors contributed to the enrichment of at least 1 functional group, while 10 LRRK2 interactors contributed to the enrichment of ≥ 12 out of the 13 functional groups (AKT1, CDK5, GSK3B, PRKCZ, HSP90AA1, MAPT, HIF1A, HDAC6, TP53, PRKN), suggesting their interactions with LRRK2 are involved in multiple biological processes. The largest numbers of LRRK2 interactors were found to contribute to the functional groups of: "intracellular organisation" (n = 179/302 interactors, 59.3%), "transport" (n = 113/302 interactors, 37.4%) "metabolism" (n = 120/302 interactors, 39.4%), "development" (n = 147/302 interactors, 48.6%), "protein modification" (n = 112/302 interactors, 37.1%), "response to stress" (n = 115/302 interactors, 38.1%), "signalling" (n = 120/302 interactors, 39.7%) and "protein catabolism" (n = 112/302 interactors, 37.1%), confirming the observation already made for the most represented functional groups (**Fig B in S1 Fig**).

Multiple Correspondence Analysis (MCA) was performed to compare the composition (in terms of LRRK2 interactors) of the 13 functional groups (**Fig B in S1 Fig**). The top three components (Dim.1, Dim.2, Dim.3) accounted for 49% of total variability in the data (23.5%, 15.0% and 10.5%, respectively). With the exception of "metabolism", "regulation of gene expression", "protein modification" and "protein catabolism" that are very close in the MCA space (this might be due to similarity in the ontology behind these terms); no other cluster was

confidently identified. This suggested the biological processes associate with LRRK2 and represented by the different functional blocks are probably sustained by distinct sets of LRRK2 interactors.

## Differential expression analysis of the general LRRK2int

RNA-Seq read counts data of LRRK2 interactors for 11 brain regions and 4 peripheral tissues were extracted from the GTEx database. Five interactors were not found in GTEx Ensembl Gene ID list and 17 presented missing expression values in $\geq 1$ tissue. These 22 interactors were thereby excluded from further analysis. Furthermore, 7 LRRK2 interactors were removed due to high missingness (having more than 5% of samples with null read counts in $\geq 1$ tissue) leaving a final total of 378 LRRK2 interactors out of 407 (92.9%) with read counts for further analysis. Of note, no outlier sample was found in any tissue, suggesting that GTEx RNA-Seq data are barely affected by batch effects (**Table H in S1 Table**).

Pair-wise differential expression analyses (DEA) were performed for each LRRK2 interactor across different tissues. Tissues were ranked based on the DEA results of each LRRK2 interactors (**Table I in S1 Table**). We were able to observe the following tissue-specific expression patterns of the general LRRK2$_{int}$: i) the tissues with the largest numbers of LRRK2 interactors showing significantly high expression levels were blood (with 162 LRRK2 interactors expressed significantly higher in blood than in $\geq 10$ of the other tissues analysed [$Score_{I_n,blood}^{Ex} \geq 10$]), cerebellum, spinal cord c-1 and frontal cortex (with n = 117; n = 74; and n = 67 highly expressed LRRK2 interactors respectively, all with $Score_{I,tissue}^{Ex} \geq 10$). The expression levels of LRRK2 interactors were generally lower in caudate, putamen, kidney cortex and liver (scored $< 9$ in 98.1% of all interactors) (**Fig 3A**). When considering the individual components of the LRRK2$_{int}$, a total of 184 out of 378 interactors (48.7%) presented with unique and significantly high expression in a certain tissue (with a $Score_{I,A}^{Ex} > 12$, suggesting that the expression level of Interactor $I$ is significantly higher in Tissue $A$ than in $\geq 12$ other tissues analysed). These 184 highly expressed interactors were distributed across 6 tissues (cerebellum, frontal cortex, spinal cord c-1, hypothalamus, anterior cingulate cortex and blood) and showed high tissue-specificity, i.e. they were highly expressed only in 1 tissue (**Fig 3B**).

A heatmap was generated based on the normalised expression matrix derived from DEA (Heatmap_DEA, **Fig 3C**). Two dendrograms were extracted from Heatmap_DEA: 1) Den_-DEA1 for the hierarchical clustering of tissues based on the overall LRRK2$_{int}$ expression patterns (**Fig C in S1 Fig**); 2) Den_DEA2 for the hierarchical clustering of each LRRK2 interactor based on expression behaviours across different tissues (**Fig D in S1 Fig**). In Den_DEA1, brain regions and peripheral tissues presented in two distinct groups, suggesting that the overall expression levels for the components of the LRRK2$_{int}$ are different in the brain in comparison with other tissues. Among the 11 brain regions, putamen, caudate, and nucleus accumbens were clustered together, indicating that the LRRK2 interactors exhibited similar expression patterns in these 3 brain regions (in terms of absolute expression levels). Of note, putamen, caudate and nucleus accumbens regions are the fundamental parts of striatum, which is substantially impacted with PD pathology. Den_DEA2 was cut following the principle of obtaining the largest number of clusters while avoiding generating clusters comprising single interactors, thereby 4 clusters were obtained (Cluster 1–4; n = 25, 124, 16 and 212 interactors, respectively, **Table J in S1 Table**). A total of 124 interactors presented in the same cluster as LRRK2 (Cluster 1, hereby referred as DEA_Cluster$_{LRRK2}$), suggesting these proteins exhibited a similar expression pattern (in terms of absolute expression levels) when compared with LRRK2 in different tissues. The 124 interactors within the DEA Cluster$_{LRRK2}$ were functionally annotated and the top significant GO:BPs based on term size $\leq 80$ showed involvement of the DEA_Cluster$_{LRRK2}$ in regulation of apoptosis and vesicle trafficking (**Table 1**).

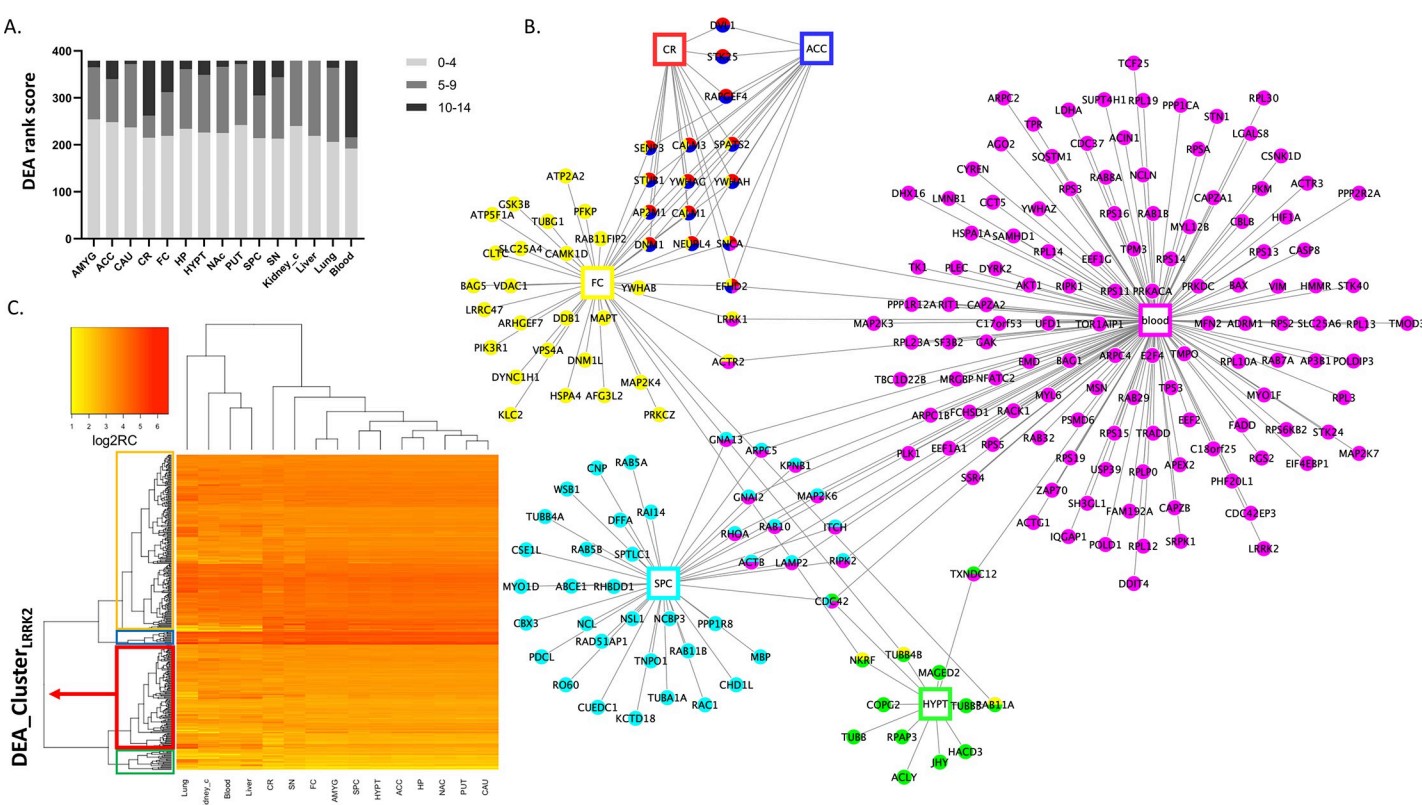

**Fig 3. Differential Expression Analysis (DEA) on the LRRK2int.** A) DEA was performed to compare the expression levels of each LRRK2 interactor across different tissues. Tissues were ranked based on the significant comparison results. The bar graph shows the distribution of the ranks of different tissues. Each bar in the graph represents a tissue, and segments in the bar represent ranks of that tissue at three levels: AVERAGE ($\leq 4$); MIDDLE (between 5 and 9); HIGH ($\geq 10$). Of note, an AVERAGE rank suggests that for interactor I, the expression level in a given tissue is lower/not significantly higher than in other tissues, or only higher than in $\leq 4$ tissues ($< 27\%$ of all tissues), while a HIGH rank means the expression level of interactor I is significantly higher in a given tissue than in $\geq 10$ other tissues ($> 67\%$ of all tissues). B) The network graph shows the LRRK2 interactors with significantly high expression in certain tissues (tissue ranks $\geq 12$), suggesting that the expression levels of these interactors in a specific tissue are significantly higher than in $\geq 12$ other tissues (86% of all tissues). Tissues are represented as rectangular nodes, while interactors are represented as round nodes. Different colour indicates different tissues. C) Heatmap_DEA was generated from normalised read counts (log2 transformed) of LRRK2 interactors in different tissues derived from DEA. Darker colour represents higher expression levels. The horizonal dendrogram of Heatmap_DEA was extracted as Den_DEA1. It shows the hierarchical clustering of tissues in which the LRRK2int exhibits similar expression patterns. The vertical dendrogram of Heatmap_DEA was extracted as Den_DEA2. It shows the hierarchical clustering of LRRK2 interactors based on the similarity of their expression figures across different tissues. Den_DEA2 was cut to generate 4 clusters of LRRK2 interactors (Cluster 1–4, marked in green, red, blue and yellow, respectively). The cluster containing LRRK2 (marked in red) is defined as DEA_ClusterLRRK2, in which the interactors presented similar overall expression distribution across tissues as LRRK2. Abbreviations: ACC: Anterior Cingulate Cortex; AMYG: Amygdala; CAU: caudate; CR: cerebellum; FC: frontal cortex; HP: hippocampus; HYPT: hypothalamus; NAc: nucleus accumbens; PUT: putamen; SN: substantia nigra; SPC: spinal cord c-1; Kidney_c: kidney cortex.

## Co-expression analysis of the general LRRK2int

Pearson's correlation test was performed to calculate the co-expression coefficients between LRRK2 and each of its 379 interactors in different tissues. The distribution of the 378 co-

**Table 1. Top terms in the GO:BP functional enrichment of the DEA ClusterLRRK2.**

| Source | Term Name | ID | adj p-value | Term size |
|--------|-----------|-----|-------------|-----------|
| GO:BP | regulation of apoptotic DNA fragmentation | GO:1902510 | 3.43E-02 | 10 |
| GO:BP | vesicle transport along actin filament | GO:0030050 | 4.62E-03 | 18 |
| GO:BP | actin filament-based transport | GO:0099515 | 7.24E-03 | 20 |
| GO:BP | positive regulation of neuron apoptotic process | GO:0043525 | 1.50E-02 | 50 |
| GO:BP | vesicle cytoskeletal trafficking | GO:0099518 | 4.54E-03 | 70 |

expression coefficients for each tissue was compared to the corresponding reference distribution generated from co-expression analysis between LRRK2 and 1000 sets of randomly picked genes (for each random gene list, n = 378 to match with the dimension of the general $LRRK2_{int}$). In frontal cortex, putamen, nucleus accumbens, hypothalamus, anterior cingulate cortex, caudate, and cerebellum, $LRRK2_{int}$ presented a larger distribution of high co-expression coefficients (0.65 to 0.85) in comparison to the reference, indicating that LRRK2 interactors were strongly correlated with LRRK2 in comparison with randomly picked genes. In hippocampus, spinal cord c-1 and lung, the LRRK2 interactors presented more moderate co-expression with LRRK2 (0.45 to 0.65) in comparison to the reference, suggesting they are mildly correlated with LRRK2 in comparison with randomly picked genes. The distribution of co-expression coefficients of the $LRRK2_{int}$ overlapped with the reference for amygdala, substantia nigra, blood, liver and kidney cortex, suggesting no significant co-expression between LRRK2 and its interactors was present in these tissues (**Fig E in S1 Fig**).

One-way ANOVA followed by Tukey's test was performed to compare the distribution of co-expression coefficients across tissues. The tissues were then ranked based on the significant results. Tissues with the higher co-expression coefficients between LRRK2 and its interactors were putamen ($Score_{PUT}^{coex}$ = 14, suggesting that the co-expression coefficients were significantly higher in putamen compared to all other 14 tissues analysed) and nucleus accumbens ($Score_{NAc}^{coex}$ = 13), followed by caudate and hypothalamus ($Score_{CAU}^{coex}$, $Score_{HYPT}^{coex}$ = 10). Hippocampus was the brain region with the lowest number of interactors co-expressed with LRRK2 ($Score_{HP}^{coex}$ = 1). Kidney cortex obtained the highest rank among the peripheral tissues ($Score_{kidney}^{coex}$ = 5) (**Fig 4A**).

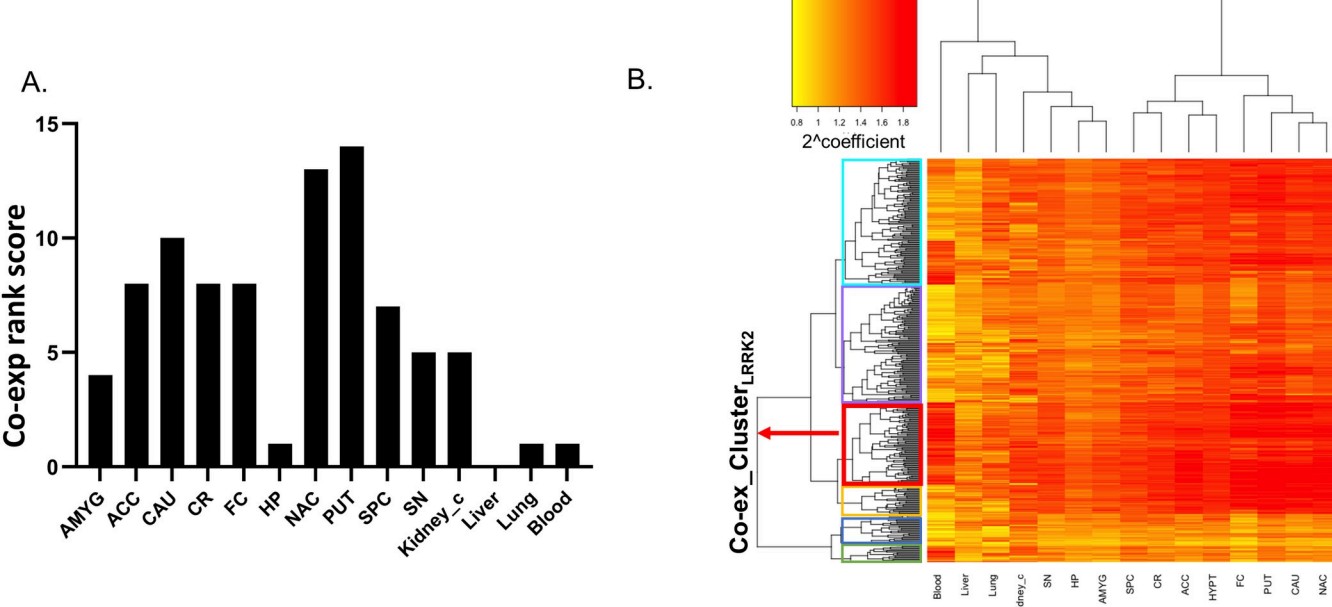

**Fig 4. Co-expression analysis on the LRRK2_int.** A) Pair-wise Tukey's test was performed to compare the co-expression coefficients (interactors vs LRRK2) across different tissues. Tissues were ranked according to the results. The bar graph shows that putamen, nucleus accumbens, caudate and hypothalamus are tissues with the highest ranks. Liver presents a rank of 0, meaning the co-expression coefficients of LRRK2 interactors are the lowest in comparison with any other tissues analysed. B) The heatmap was generated from the coefficient matrix (2^coefficient = 2 raised to the power of the coefficient) derived from the co-expression analysis (Heatmap_Co-ex). Darker colour represents higher co-expression coefficient. The horizonal dendrogram of Heatmap_Co-ex was extracted as Den_Co-ex1, which shows the hierarchical clustering of tissues in which the LRRK2 interactors exhibited similar co-expression patterns with LRRK2. The vertical dendrogram of Heatmap_Co-ex was extracted as Den_Co-ex2, which shows the hierarchical clustering of interactors based on the similarity of their co-expression figures with LRRK2 across different tissues. Den_Co-ex2 was cut to generate 6 clusters of LRRK2 interactors (Cluster A-F, marked in green, blue, yellow, red, purple and turquoise, respectively). Interactors in Cluster D presents the highest level of overall co-expression behaviour with LRRK2 across different tissues (referred as Co-ex_Cluster_LRRK2). Abbreviations: ACC: Anterior Cingulate Cortex; AMYG: Amygdala; CAU: caudate; CR: cerebellum; FC: frontal cortex; HP: hippocampus; HYPT: hypothalamus; NAc: nucleus accumbens; PUT: putamen; SN: substantia nigra; SPC: spinal cord c-1; Kidney_c: kidney cortex.

A heatmap was generated from the coefficient matrix derived from co-expression analysis (Heatmap_Co-ex, Fig 4B). The 2 dendrograms in Heatmap_DEA were extracted as follows: 1) Den_Co-ex1 for the hierarchical clustering of tissues in which the LRRK2$_{int}$ presented similar co-expression patterns (Fig F in S1 Fig); 2) Den_Co-ex2 for the hierarchical clustering of the LRRK2 interactors that exhibited similar co-expression behaviours with LRRK2 across different tissues (Fig G in S1 Fig). In Den_Co-ex1, two clusters were identified: i) frontal cortex, putamen, nucleus accumbens and caudate; ii) spinal cord c-1, hypothalamus, cerebellum and anterior cingulate cortex, indicating that the pattern of co-expression between LRRK2 and its interactors was different between the 2 clusters but similar for the tissues within each cluster. This suggests that the LRRK2$_{int}$ may participate in different cellular functions in the 2 clusters of brain regions.

A cut-off was applied to the Den_Co-ex2 to cluster LRRK2 interactors based on the similarity of co-expression behaviour (with LRRK2) across different tissues. The cut-off was set to obtain the largest number of clusters while avoiding generating clusters comprising single interactors. A total of 6 clusters was generated (Cluster A-F; n = 15, 30, 27, 77, 110, 118, respectively), in which Cluster D (n = 77/378, 20.3%) contained the LRRK2 interactors presenting the highest overall co-expression coefficients with LRRK2 across different tissues (Table K in S1 Table). Cluster D represents the group of LRRK2 interactors whose genes are more frequently co-expressed with LRRK2 considering the pool of tissues under investigation (hereby referred as Co-ex_Cluster$_{LRRK2}$).

The 77 interactors within the Co-ex_Cluster$_{LRRK2}$ were functionally annotated and the top significant GO:BPs based on term size $\leq$ 80 were different from those identified with the DEA Cluster$_{LRRK2}$ showing involvement of the Co-ex_Cluster$_{LRRK2}$ in autophagy, actin metabolism and lamellipodium assembly (Table 2).

## Tissue specific LRRK2 interactomes

The DEA rank (Table I in S1 Table) indicates, for a given LRRK2 interactor in a given tissue, the number of other tissues (out of 15) in which the expression of that specific LRRK2 interactor is significantly lower in comparison with the tissue under analysis. Therefore, the DEA ranks represent a way to identify LRRK2 interactors that are differentially expressed across tissues as these will receive largest ranks in comparison with interactors that have a constant level of expression across tissues.

Table 2. Top terms in the GO:BP functional enrichment of the Co-ex_Cluster$_{LRRK2}$.

| Source | Term Name | ID | adj p-value | Term size |
|--------|-----------|-----|-------------|-----------|
| GO:BP | positive regulation of autophagic cell death | GO:1904094 | 2.40E-02 | 2 |
| GO:BP | chromosome movement towards spindle pole | GO:0051305 | 2.87E-03 | 7 |
| GO:BP | amyloid-beta clearance by transcytosis | GO:0150093 | 2.87E-03 | 7 |
| GO:BP | positive regulation of lamellipodium assembly | GO:0010592 | 1.06E-04 | 29 |
| GO:BP | positive regulation of lamellipodium organization | GO:1902745 | 3.80E-04 | 37 |
| GO:BP | Arp2/3 complex-mediated actin nucleation | GO:0034314 | 2.12E-02 | 39 |
| GO:BP | regulation of lamellipodium assembly | GO:0010591 | 1.29E-05 | 41 |
| GO:BP | actin filament capping | GO:0051693 | 3.45E-02 | 44 |
| GO:BP | negative regulation of actin filament depolymerization | GO:0030835 | 4.89E-02 | 48 |
| GO:BP | regulation of lamellipodium organization | GO:1902743 | 6.36E-05 | 53 |
| GO:BP | lamellipodium assembly | GO:0030032 | 4.46E-04 | 73 |
| GO:BP | positive regulation of cell morphogenesis involved in differentiation | GO:0010770 | 1.86E-02 | 80 |

Similarly, the LRRK2:interactors co-expression coefficients represent a way to identify LRRK2 interactors that are significantly co-expressed with LRRK2 in different tissues as these will score higher than interactors that do not have a significant co-expression with LRRK2.

The DEA ranks and the co-expression coefficients were combined in **Table L in S1 Table**; this table can be filtered (with custom made thresholds) to generate tissue-specific LRRK2$_{ints}$. This table can be also used to evaluate how, a given LRRK2 interactor, is differentially expressed or co-expressed across the 15 different tissues. As an example of how this table can be used to generate tissue-specific LRRK2 interactomes and how it can help in generating testable hypotheses, we have conducted a comparison of the LRRK2 interactome in the striatal functional unit that we previously identified vs the cerebellum.

Firstly, we plotted each of the LRRK2 interactors highlighting the different distribution of the DEA ranks and the co:expression coefficients across the 4 different tissues (**Fig 5A**). We then extracted the tissue-specific LRRK2$_{ints}$ for these 4 tissues by setting a DEA threshold as rank > 8 and co-expression threshold as co-expression coefficient $\geq$ 0.7 (**Table M in S1 Table**) and keeping, for each brain region, only the LRRK2 interactors above these thresholds (tissue-specific interactors).

In putamen, caudate and nucleus accumbens, the core LRRK2 interactors presented low/ moderate DEA ranks, indicating they are not particularly expressed in these 3 tissues in comparison with the others. In fact, the amount of tissue-specific LRRK2 interactors in the putamen, caudate and nucleus accumbens with a DEA rank > 8 accounted for only the 3.9, 4.5 and 7.1% of the entire LRRK2$_{int}$(**Fig 5B**). However, high co-expression levels with LRRK2 were observed in these 3 tissues. In fact, the amount of tissue-specific LRRK2 interactors in the

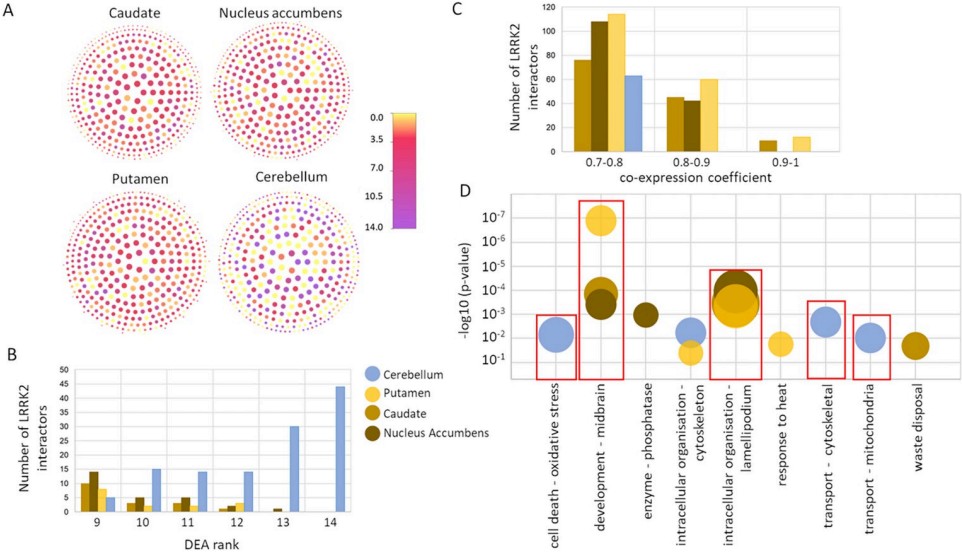

**Fig 5. Tissue specific LRRK2$_{ints}$.** A) The network graphs represent the LRRK2$_{ints}$ with nodes color-coded for DEA ranks and dimension-coded based on the co-expression coefficients. The darker the colour, the higher the expression level of the interactor in a given tissue. The larger the node, the higher the co-expression coefficient calculated between LRRK2 and the interactor in a given tissue. B) For each of the 4 brain regions the LRRK2 interactors with a DEA rank > 8 were extracted and the distribution of the DEA ranks was visualized. C) For each of the 4 brain regions the LRRK2 interactors with a co-expression coefficient $\geq$ 0.7 were extracted and the distribution of the co-expression coefficients was visualized. D) Functional enrichment results for the tissue-specific LRRK2 interactors; GO:BPs were filtered for term size $\leq$ 80 to keep only specific terms and GO:BPs were grouped based on semantic similarity. The semantic class is reported on the x-axis, the p-value of the most significant term in the semantic class is reported on the y-axis, the dimension of the circle represent the number of GO:BPs terms grouped in the semantic class. For simplicity, the semantic classes containing only 1 GO:BP term were omitted from the graph. -log10(p-value) = p-value in logarithmic scale, base 10, reverse order.

putamen, caudate and nucleus accumbens with a co-expression coefficient $\geq 0.7$ accounted for the 49.2, 34.4 and 39.7% of the entire LRRK2$_{int}$(**Fig 5C**).

In the cerebellum the contrary was observed: the tissue-specific LRRK2 interactors had high DEA ranks (the amount of tissue-specific LRRK2 interactors with a DEA rank > 8 accounted for 32% of the entire LRRK2$_{int}$ in the cerebellum); however, the LRRK2:interactors co-expression was generally lower as compared to putamen, caudate and nucleus accumbens (the amount of tissue-specific LRRK2 interactors with a co-expression coefficient $\geq 0.7$ accounted for only 16.7% of the entire LRRK2$_{int}$ in the cerebellum) (**Fig 5B and 5C**).

As LRRK2 functions depend on which interactors are available in a certain tissue to perform a biological action, these results might be an indication that the LRRK2 functions in putamen, caudate and nucleus accumbens are substantially different from those in the cerebellum.

We therefore used the tissue-specific interactors extracted for each of the 4 brain regions (DEA rank > 8 U co-expression coefficient $\geq 0.7$) to run functional enrichment for GO:BPs (**Table N in** S1 Table). After QC, performed to remove the more general terms that generate background noise in the analysis, and grouping of the GO:BPs based on semantic similarity into broader classes, we compared the results of the functional enrichment (**Fig 5D**). Strikingly, the tissue-specific LRRK2 interactors in putamen, caudate and nucleus accumbens showed a specific enrichment of terms related to development of the midbrain and organization of the lamellipodium. In the cerebellum these terms were missing while there was specific enrichment for terms related with cell-death associated with oxidative stress, and transport related to the cytoskeleton and the mitochondria.

## Differentiating LRRK2:Rab interactions in the CNS and the periphery

Table L in S1 Table can be used to explore specific interactors of interest; here we selected the Rab proteins as an example. A total of 13 Rab proteins were identified in the general LRRK2$_{int}$ (RAB38, RAB10, RAB11A, RAB11B, RAB11FIP2, RAB1A, RAB1B, RAB29, RAB32, RAB5A, RAB5B, RAB7A and RAB8A), while RAB38 was excluded due to its incomplete expression data across 15 tissues of analysis, thereby only 12 Rab proteins were included for further analysis. Of note, among these Rab proteins, RAB29, RAB8A and RAB11FIP2 presented in the DEA_Cluster$_{LRRK2}$ (**Table J in** S1 Table), suggesting these 3 proteins have a similar expression pattern as LRRK2 across all the 15 tissues analysed; while RAB10, RAB11A, RAB11FIP2, RAB1A, RAB1B, RAB5A, RAB5B, RAB7A and RAB8A presented in the Co-ex_Cluster$_{LRRK2}$ (**Table K in** S1 Table), indicating that these Rab proteins show a generally higher co-expression with LRRK2 in all the 15 tissues analysed.

In the functional enrichment of the general LRRK2$_{int}$ all of the 12 Rab proteins contributed to the enrichment of the functional group "transport" together with LRRK2, while 11 of them were part of the group "intracellular organisation", suggesting that these 2 functions are generally sustained by LRRK2-Rab cooperation (**Fig 6**) independently of the identity of the Rab protein. Five Rab proteins (RAB1A, RAB1B, RAB5A, RAB7A and RAB8A) presented in the "autophagy" functional group, potentially indicating they participate in the waste-disposal processes regulated via LRRK2.

After filtering out non-Rab proteins from the LRRK2$_{ints}$ for each tissue, 15 tissue specific LRRK2:Rab interactomes were generated (**Fig 7**). Among all Rab proteins, RAB7A and RAB5B presented high expression levels across the majority of the tissues analysed (i.e. absolute expression level of RAB7A and RAB5B > [Mean of other Rab proteins in the tissue] +1SD). The tissues with the highest DEA score were spinal cord and substantia nigra (DEA rank score = 14 and 10, respectively) meaning RAB7A and RAB5B levels were significantly higher in spinal cord and substantia nigra than in 14/14 and 10/14 other tissues. RAB10,

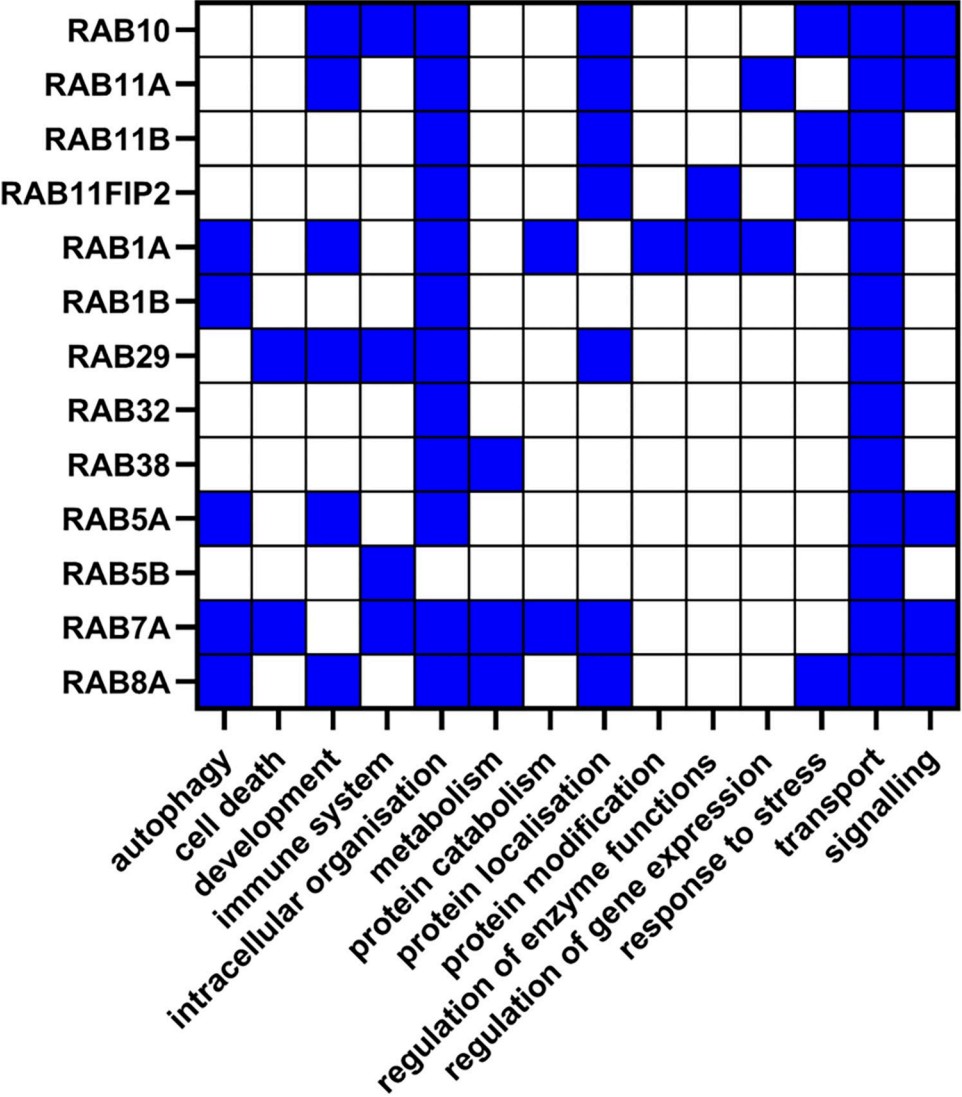

**Fig 6. Functional roles of Rab interactors of LRRK2.** The heatmap shows the functional groups that included the Rab proteins presented in the LRRK2int. Blue squares represent the presence of a certain Rab interactor in a given functional group identified in the functional enrichment analysis for the general LRRK2int.

RAB11FIP2, RAB11A, RAB7A and RAB5B exhibited the highest co-expression with LRRK2 in the majority of the tissues. Among the brain regions, putamen, caudate, frontal cortex and nucleus accumbens showed identical distribution of absolute expression levels of Rab interactors and similarly high co-expression between these proteins and LRRK2. Of note, RAB32 and RAB29 presented a similar co-expression pattern with LRRK2 across different tissues as compared to other Rab proteins. A high LRRK2:RAB32 correlation was seen in hypothalamus, cerebellum, substantial nigra spinal cord c-1 and blood. Of note, RAB32 presented the highest expression level in blood ($Score^{Ex}_{RAB32,blood} = 14$), suggesting a potentially more important role of LRRK2:RAB32 interaction in blood. Similarly, RAB29 presented a high co-expression with LRRK2 in hypothalamus, substantia nigra and spinal cord c-1. These suggest a potential co-function among LRRK2, RAB32 and RAB29 in these brain regions. In comparison, LRRK2:

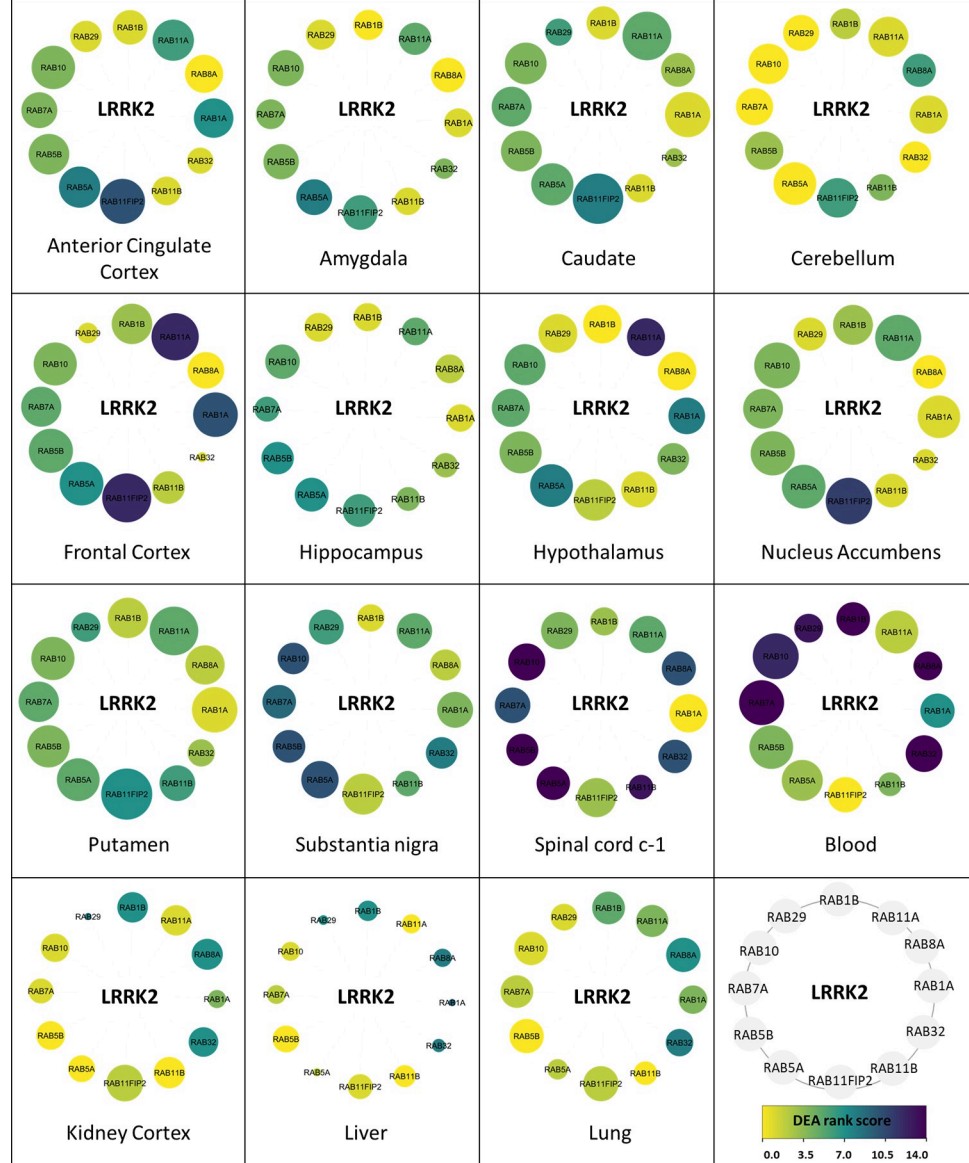

**Fig 7. Tissue specific LRRK2:Rab interactomes (LRRK2ints_Rab).** The network graphs show the different expression attributes of the 12 Rab proteins in the general LRRK2int. Nodes represent Rab interactors of LRRK2 (in grey is represented the scheme with the order of the individual Rab proteins in the graphs). Node colour represents the DEA rank scores of Rab proteins. The darker the colour, the higher the DEA rank of a certain Rab interactor of LRRK2 for a given tissue. Node size represents co-expression coefficients. The larger the node, the higher the co-expression coefficient between LRRK2 and the Rab protein in a given tissue.

Rab co-expression was weak in amygdala and lung, while the poorest co-expression was seen in hippocampus, liver and kidney cortex. The co-expression of LRRK2 and RAB29 in the substantia nigra is of particular interest, as RAB29 is a candidate risk gene for Parkinson's [43], is a putative substrate for LRRK2 kinase activity [44], and may regulate LRRK2 [45,46]. It is also of note that only a subset of the Rab proteins reported as interactors for LRRK2 have been validated as substrates for LRRK2 kinase activity.

## Discussion

In this study, we constructed the LRRK2 protein-protein interactome from PPIs derived from peer-reviewed literature. To maximise literature coverage, PPIs were obtained from 3 secondary repositories: HIPPIE, MIST and PINOT. These tools extract PPIs from multiple primary, manually curated databases (**Fig H in S1 Fig**) allowing to effectively retrieve the available PPI literature. We removed all scoring filters while downloading interactions. This allowed us to retrieve as many LRRK2 interactions as possible from the literature. Therefore, to the best of our knowledge, the interactome presented in this manuscript represents the most comprehensive reconstruction of the LRRK2 interactome to date (1871 LRRK2 interactors).

With such a large number of LRRK2 interactors, it was important to set filters to extract the most reliable core within the vast LRRK2 interactome. We defined "reliability" as "reproducibility", therefore we reduced the LRRK2 PPIs to those that have been reported at least twice in literature. To achieve this, we followed a 2-step strategy. Firstly, we converted similar interaction detection methods into one single method class (using the PINOT Method Grouping Dictionary, **S1 Table**). This step was essential for accurate evaluations of the reproducibility of the LRRK2 PPIs, as curators behind different databases might record the same PPI under multiple interaction detection methods that are actually methodological synonyms, thereby introducing semantic bias. Secondly, "Low quality" PPIs–defined as interactions reported in literature within 1 publication only and validated with methods belonging to 1 category only–were filtered out from the LRRK2 interactome.

Following the 2-step strategy, the LRRK2 interactome was reduced to only 407 out of 1871, clearly showing that one of the problems with LRRK2 investigations (and potentially with PPI analyses in general) is that most of the data are not reproduced and thereby not directly trustable. This poor reproducibility may result from delays in curation of papers into primary databases, limited interest in wet-lab research aimed at reproducing information that has already been published, or the real inflation in type 1 errors.

The 407 interactors that passed the quality control constituted the general LRRK2 interactome, which is to our knowledge the most comprehensive, quality controlled human protein interactome of LRRK2 (LRRK2$_{int}$) currently available.

It is important to consider, however, that PPI analyses suffer from ascertainment bias as more studied and more abundant proteins tend to be more represented. Additionally, there is no tissue/cell specific dimension to the PPI data as protein interactions from experimental data tend to be generated in model systems (i.e. HEK293 cells) or completely in vitro systems (proteo-arrays).

Further analysis of the general LRRK2 interactome was performed to evaluate whether the approach adopted in the construction of the general LRRK2 interactome was adequate to capture the current knowledge around LRRK2. Among the 407 interactors, LRRK2 itself exhibited the highest score for interaction (FS = 50), confirming LRRK2 self-interaction as the best known and reproducible PPI of LRRK2. Heterologous interactors with high FS (i.e. replicated in many publications using a plethora of different methods) were, as expected: HSP90AA1 (FS = 19); YWHAQ (FS = 14), HSPA8 (FS = 13), MSN (FS = 13), YWHAZ (FS = 13), CDC37 (FS = 11), DNM1L (FS = 11), STUB1 (FS = 11) and TUBB (FS = 11). A large proportion of the general LRRK2$_{int}$ was composed of cytoskeletal proteins (12.5%), which is in accordance with the known role of LRRK2 in cytoskeletal dynamics (reviewed by [47]). Finally, 35 protein kinases and 13 Rab GTPases were part of the LRRK2$_{int}$, confirming, again, that the general LRRK2$_{int}$ built in our analysis does recapitulate the general knowledge related to LRRK2 interaction partners.

Functional enrichment analysis of the general LRRK2$_{int}$ identified a plethora of biological processes, which were grouped into 13 larger functional groups. MCA was performed to

compare the contribution of each single LRRK2 interactor to the enrichment of these functional groups. Interestingly, the majority of the functional groups were not combined in any cluster. This result confirmed that i) the LRRK2$_{int}$ is indeed involved in a wide range of different functions and ii) when looking at each of these functions, it appears that they are driven by distinct sets of LRRK2 interactors. In other words, LRRK2 interactors do not contribute, as a group, to the enrichment of all the LRRK2 associated functions. Rather, specific sets of LRRK2 interactors contribute to the enrichment of specific LRRK2 associated functions. This note is of importance as it sustains the hypothesis that differential expression of groups of LRRK2 interactors in different tissues might indeed be responsible for the specialisation of LRRK2 functions. In this model, the multiple functions the LRRK2$_{int}$ is associated with are not all active at the same time/place; rather they are differentially relevant within different tissues, cell types and moments in time.

RNA-seq read counts for the LRRK2 interactors were downloaded from GTEx for 11 brain regions and 4 peripheral tissues. These data were used to differentiate the general LRRK2$_{int}$ into 15 tissue specific LRRK2 interactomes. Two different strategies were applied based on pair-wise DEA (considering absolute expression levels) and pair-wise co-expression analysis (considering LRRK2:interactor co-expression levels). In general, the level of co-expression is regarded as a preferential measure as it is assumed that co-expressed proteins are co-regulated at a gene level and involved in the same biological processes. However, there is no reason to rule out an analysis based on DEA as an increased level of expression for a certain protein in a tissue might indicate increased demand for the biological processes in which that protein participate. Therefore, both DEA and co-expression analysis were used to analyse the expression behaviour of the LRRK2 interactors in 15 different tissues following the assumption that LRRK2 interactors can increase or decrease in concentration in a tissue specific fashion following the unique functional requirements of each tissue. Similarly, in different tissues, LRRK2 can be co-expressed with different interactors depending on the functional unit that need to be activated to sustain the tissue specific functions.

We then used these results to compare the 15 tissues against each other. Firstly, we evaluated the hierarchical clustering of the 15 tissues to compare the expression behaviour of the LRRK2$_{ints}$ in the 4 peripheral tissues vs the 11 brain regions. The results showed a clear difference in expression profiles (peripheral tissues vs CNS). In the DEA analysis the 4 peripheral tissues formed a distinct cluster meaning the LRRK2 interactors are in general expressed at different levels in the periphery in comparison with the CNS. In the co-expression analysis, the different peripheral tissues did not cluster together, however, they did not cluster with the brain regions either. In particular, when the brain regions were all already grouped into 3 clusters, the peripheral tissues were still left unclustered. This might suggest that from a co-expression perspective LRRK2 interactors behave similarly in some brain regions, while they show peculiar and unique behaviours in the 4 peripheral tissues.

Interestingly, the hierarchical clustering of the 11 brain regions based on the results from both DEA and co-expression analysis showed a similar result: putamen, caudate and nucleus accumbens were always found within the same cluster, suggesting the expression behaviour of the LRRK2$_{int}$ is very similar within these 3 regions both in terms of absolute expression levels of LRRK2 partners and LRRK2:interactors co-expression. This suggests the 3 regions might form a LRRK2 functional unit within the brain where the LRRK2$_{int}$ sustains similar processes/functions.

Of note, putamen, caudate and nucleus accumbens form the striatum, target for the projection of dopaminergic neurons and one of the most affected brain regions during Parkinson's disease (PD) progression. In fact, multiple studies have associated the degeneration of putamen and caudate with motor and non-motor PD symptoms [48–50]; while nucleus

accumbens, involved in mediating emotional and motivational processes such as rewarding experiences, impulsive and compulsive behaviours, might be implicated in the neuropsychiatric symptoms of PD [51,52].

The results from the expression analyses were also used to compare and rank the LRRK2 interactors based on their expression behaviour across the 15 different tissues. A total of 124 interactors presented in the same differential expression cluster as LRRK2 (named DEA_Cluster$_{LRRK2}$). This cluster represents the proteins within the LRRK2$_{int}$ that share the same pattern of differential expression as LRRK2 in the majority of tissues under analysis. A smaller number of interactors (n = 77) were grouped in the Co-ex_Cluster$_{LRRK2}$, which is composed of the proteins (within the LRRK2$_{int}$) presenting with the highest co-expression level with LRRK2 in the majority of the analysed tissues. Of note, 30 interactors overlapped between these two clusters, meaning they showed both conserved co-expression with LRRK2 and similar expression profiles as LRRK2 across the majority of tissues. Due to this peculiar behaviour, these 30 interactors might be the gateway to understand the "constitutive" LRRK2 functions that are conserved in different districts.

We have presented Table L in S1 Table as a resource that can be used to filter by differential expression levels and by co-expression levels the general LRRK2$_{int}$ thus obtaining tissue-specific LRRK2$_{ints}$. Unfortunately, thresholds are arbitrary, however they allow to identify the core LRRK2 interactors in a given tissue based on their expression behaviour. We have exemplified the use of Table L in S1 Table to generate tissue-specific LRRK2$_{int}$ firstly by comparing the expression behaviour of LRRK2 interactors in putamen, caudate and nucleus accumbens (the striatal unit) in comparison with cerebellum. Functional enrichment of the single tissue-specific LRRK2$_{int}$ confirmed the striatal tissues as a functional unit as no differences were identified across the functions sustained by LRRK2 in the 3 regions. However, clear differences were observed when the whole striatum was compared with the cerebellum, suggesting that (at least in our model) the functions of LRRK2 exerted via its interactome are different in these 2 brain regions.

Finally, we complemented our analysis by investigating the Rab protein family as an example to show how Table L in S1 Table can be used to answer a specific research question regarding a singular or a particular group of interactors of interest. We extracted information focused on the LRRK2 interactors belonging to the Rab protein family thus describing the LRRK2-Rab relationship (from the perspective of expression and co-expression behaviours) across different tissues. Rab proteins have been widely recognised as LRRK2 interactors. They have been reported to cooperate with LRRK2 in a number of cellular processes such as the regulation of endolysosomal functions, response to stress and vesicle trafficking [29,53]. Here we used tissue specific LRRK2:Rab interactomes to describe how this cooperation between LRRK2 and Rab proteins may potentially vary across different tissues. Our results suggested that, in general, caudate, putamen and nucleus accumbens (that we previously defined as striatal LRRK2 functional unit) have the highest levels of LRRK2:Rab co-expression, hinting a potentially stronger association between LRRK2 and the Rab proteins in the striatum in comparison with other brain regions or peripheral tissues. Among these Rab proteins, RAB7A and RAB5B presented high co-expression levels with LRRK2 in most of the tissues and within each single tissue they showed expression levels higher than the mean + 1SD of all other Rab proteins, suggesting these 2 Rab proteins may play a constitutive and conserved role in the LRRK2:Rab interactome across the whole body. In comparison, RAB32 and RAB29 presented unique LRRK2:co-expression patterns with higher co-expression levels in substantia nigra, hypothalamus, spinal cord c-1, cerebellum and blood, suggesting these 2 Rab proteins may be specific LRRK2 partners in these tissues only.

## Conclusion

PPIs can help in understanding the functional milieu around a hub protein, such as LRRK2 in this study. However, tissue specificity is generally not considered in these analyses. With this work we have designed a pipeline that makes use of expression data to provide indication of tissue specific differences within the interactome of a given hub protein (LRRK2 in this case). On one hand we provided evidence that brain tissues are different from peripheral tissues concerning expression patterns of the LRRK2$_{int}$ and within the brain we defined a cluster composed of caudate, putamen and nucleus accumbens, where the LRRK2 interactors shows very conserved expression patterns. Additionally, we identified 30 LRRK2 interactors, showing the most conserved co-expression with LRRK2 and the most similar expression profile as LRRK2 across the majority of tissues analysed. On the other hand, we presented examples to illustrate how the data generated via our pipeline (available for download) can be filtered for answering research questions regarding the tissue specificity of LRRK2 PPIs. As LRRK2 is a crucial target for PD treatment, with several small molecules currently in clinical trials, a better understanding of LRRK2's tissue specific functionality has indeed become a research priority. This pipeline intends to be a rigorous bioinformatical attempt to raise awareness of the complexity and variability of LRRK2's interplay with its interactors.

## Supporting information

**S1 Fig. Fig A in S1 Fig.** Overlap of LRRK2 interactors downloaded from PINOT, HIPPIE and MIST. **Fig B in S1 Fig.** Functions of LRRK2int A) The bar graph shows the numbers of interactors contributing to the enrichment of different functional groups. B) The 14 functional groups were then analysed via MCA to compare their composition in terms of LRRK2 interactors. **Fig C in S1 Fig.** Den_DEA1 presents the hierarchical clustering of tissues in terms of the overall expression pattern of the LRRK2int. **Fig D in S1 Fig.** Den_DEA2 presents the hierarchical clustering of LRRK2 interactors based on the expression patterns across different tissues (for the list of gene names refer to S10 Table). **Fig E in S1 Fig.** Co-expression coefficients distribution in 15 tissues. each tissue, the distribution of co-expression coefficients between LRRK2 and its interactors (in red) was compared to the "reference distribution" (co-expression coefficients between LRRK2 and 1000 randomly picked gene sets, in black). A: amygdala-LRRK2$_{int}$; B: anterior_cingulate_cortex-LRRK2$_{int}$; C: caudate-LRRK2$_{int}$; D: frontal_cortex-LRRK2$_{int}$; E: hippocampus-LRRK2$_{int}$; F: hypothalamus-LRRK2$_{int}$; G: nucleus_accumbens-LRRK2$_{int}$; H: putamen-LRRK2$_{int}$; I: spinal_cord_c-1-LRRK2$_{int}$; J: substantial_nigra-LRRK2$_{int}$; K: cerebellum-LRRK2$_{int}$; L: blood-LRRK2$_{int}$; M: liver-LRRK2$_{int}$; N: lung-LRRK2$_{int}$; O: kidney_cortex-LRRK2$_{int}$. **Fig F in S1 Fig.** Den_Co-ex1 presents the hierarchical clustering of tissues in terms of the co-expression pattern between LRRK2 and its interactors. **Fig G in S1 Fig.** Den_Co-ex2 presents the hierarchical clustering of LRRK2 interactors in terms of their overall co-expression behaviours with LRRK2 across tissues analysed (for the list of gene names refer to Table S11). **Fig H in S1 Fig.** Overlap of primary databases in PINOT, HIPPIE and MIST (PDF)

**S1 Table. Table A in S1 Table.** PINOT Method. **Table B in S1 Table.** Merged list. **Table C in S1 Table.** FS>2. **Table D in S1 Table.** family.check. **Table E in S1 Table.** Original.enrichment. **Table F in S1 Table.** GOterms(LRRK2 only). **Table G in S1 Table.** functional.groups. **Table H in S1 Table.** RNA.QC. **Table I in S1 Table.** DEA.ranks. **Table J in S1 Table.** DEA.clusters. **Table K in S1 Table.** Coex.clusters. **Table L in S1 Table** Tissue specific ints. **Table M in S1 Table.** CAU-NAC-PUT-CR. **Table N in S1 Table.** enrichment (XLSX)

## Author Contributions

**Conceptualization:** Claudia Manzoni.

**Data curation:** Yibo Zhao.

**Formal analysis:** Yibo Zhao, Claudia Manzoni.

**Supervision:** Nikoleta Vavouraki, Ruth C. Lovering, Valentina Escott-Price, Kirsten Harvey, Patrick A. Lewis, Claudia Manzoni.

**Writing – original draft:** Yibo Zhao, Claudia Manzoni.

**Writing – review & editing:** Yibo Zhao, Nikoleta Vavouraki, Ruth C. Lovering, Valentina Escott-Price, Kirsten Harvey, Patrick A. Lewis, Claudia Manzoni.

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
