## [Decision Letter · Decision Letter 0]

14 Sep 2022

Dear Dr manzoni,

Thank you very much for submitting your manuscript "Tissue specific LRRK2 interactomes reveal a distinct functional unit within the striatum" for consideration at PLOS Computational Biology.

As with all papers reviewed by the journal, your manuscript was reviewed by members of the editorial board and by two independent reviewers. In light of the reviews (below this email), we would like to invite the resubmission of a significantly-revised version that takes into account the reviewers' comments.

Both reviewers appreciated that your work is solid but have some concerns that you need to address.

We cannot make any decision about publication until we have seen the revised manuscript and your response to the reviewers' comments. Your revised manuscript is also likely to be sent to reviewers for further evaluation.

Sincerely,

Miguel A. Andrade-Navarro, Ph.D.

Guest Editor

PLOS Computational Biology

William Noble

Section Editor

PLOS Computational Biology

Both reviewers appreciated that your work is solid but have some concerns that you need to address.

Reviewer's Responses to Questions

**Comments to the Authors:**

Reviewer #1: The authors construct brain-specific interactomes around the Parkinson's disease protein LRRK2. To this end, they combine PPI data with gene expression information from different public resources. Their work is interesting yet mostly describing the different generated networks. They highlight a few interesting observations that could contribute to / explain characteristics of LRRK2 pathology. Eg. they observe similarities in different subregions of the striatum.

Overall the work is clear and does not contain any major flaws. I list a few comments below:

Major comments:

Are the results of the MCA shown in Figure 3B specific to LRRK2 interactors? Or is the clustering a more general property of those functional categories? Eg do protein catabolism and protein modification cluster also when all proteins annotated with those categories are considered? Or only for the subset of LRRK2 interactors?

Which version of the different resources did the authors use? Eg. the current version of HIPPIE lists 1,543 interactions formed by LRRK2, so I suppose the authors used a previous version. They should indicate for all databases which version they used to allow for reproducebility of the results.

To which degree are the functional enrichments a consequence of technical and research biases? I understand from the description in the Methods section that no background was used for determining the functional enrichments. However, the set of interacting proteins is already biased towards certain protein classes (eg highly expressed) due to mostly technical biases. Filtering towards interaction partners detected several times as the authors do, will likely further bias the network towards well studied proteins. The authors should test to which degree the GO term analysis results are due to those biases by testing against a suitable background (at least all proteins in the PPI networks used but even better comparing to proteins with similar network topological properties).

Minor comments:

is LRRK2-ints the same us LRRK2_ints_ (where _ denotes subscript)? In general abbreviations should be clearly defined at their first accurence in the manuscript and then consistently used.

Partly the organization of information into Methods and Results could be done better. Eg both the Results as well the Methods section have a section entitled "RNA-Seq data download and quality control" with highly redundant content. The manuscript could substantially benefit from avoiding this redundancy (in most cases I'd shorten the Results part).

"These tools extract PPIs from multiple primary, manually curated databases allowing to effectively text mine the available PPI literature -> the term "text mining" is misleading here

I find the first paragraph of the Discussion misleading: why discussing factors that distinguish the three PPI source databases used (MIST, HIPPIE and PINOT) but were not really taken into account or are relevant for the study (eg the scoring of PPIs or the download time of interactions).

Reviewer #2: By integrating data from different public databases, Zhao and colleagues provide a comprehensive and technically sound computational analysis of the tissue specificity of the LRRK2 interactome. In this work, the authors overlaid tissue specific expression with protein-protein interaction (PPI) data available in the databases. Data confidence scores for individual LRRK2 interactors have also been considered. The tissue specificity of the LRRK2 associated signaling has long been discussed in the field, especially in the context of organ specific phenotypes. For this reason, the present bioinformatic analysis may definitely contribute to a better understanding of LRRK2 signaling by systematically annotating the tissue specific expression of individual LRRK2 PPIs. This is also of particular interest for the development of specific and effective targeting strategies as it may help to understand and (even) reduce tissue specific (on-target) side effects of LRRK2-centric intervention in the future.

I nevertheless have a couple points which have to be addressed to overcome limitations of the study:

1) While the manuscript is very much focusing on technical details, the biological meaning of the results remains largely unclear. I would therefore like to encourage the authors to provide a more detailed discussion on the functional outcome of the study to help the reader to better understand the tissue specific LRRK2 function and to potentially come up with testable hypotheses.

2) If I understand it correctly, the major approach is to overlay tissue specific expression profiles with the published LRRK2 interactome. However, the experimental data might have a bias due to the over representation of some techniques and cell systems which also impacts the confidence score of individual interactors. I would appreciate if the authors could comment on how the experimental bias is considered/ compensated in the study.

3) As the readability of the figures is limited in several cases, I would recommend to increase the font size for the annotations. In particular, the informative value of Figure 6 is quite limited due to unreadable protein/ gene names.

**Have the authors made all data and (if applicable) computational code underlying the findings in their manuscript fully available?**

Reviewer #1: None

Reviewer #2: Yes

PLOS authors have the option to publish the peer review history of their article (what does this mean?). If published, this will include your full peer review and any attached files.

Reviewer #1: No

Reviewer #2: No
---

## [Decision Letter · Decision Letter 1]

3 Jan 2023

Dear Dr manzoni,

We are pleased to inform you that your manuscript 'Tissue specific LRRK2 interactomes reveal a distinct striatal functional unit' has been provisionally accepted for publication in PLOS Computational Biology.

Best regards,

Miguel A. Andrade-Navarro, Ph.D.

Guest Editor

PLOS Computational Biology

William Noble

Section Editor

PLOS Computational Biology

Reviewer's Responses to Questions

**Comments to the Authors:**

Reviewer #1: The authors have addressed most of my comments in a careful manner. I think the quality of the manuscript improved.

The only point that the authors decided not to address is my comment nr 3 (regarding ascertainment bias). Instead of performing the test I suggested (how does the functional enrichment change when considering subnetworks of similar topology, ie testing specifically the impact of well studied proteins on their GO enrichment), they repeated the analysis with a different background (expanding to all human genes). I find this control unnecessary. However, in addition the authors add some sort of disclaimer regarding the ascertainment bias to the Discussion. I think this addresses my point not optimally but sufficiently well.

I am therefore in favor of publication of this article.

Reviewer #2: Based on the reviews, the authors significantly improved the manuscript. In particular, the results section has been restructured, making the conclusions drawn much clearer. In addition, the figures have been improved.

In conclusion, all points have been adequately addressed. The manuscript can therefore be accepted in its present form.

**Have the authors made all data and (if applicable) computational code underlying the findings in their manuscript fully available?**

Reviewer #1: Yes

Reviewer #2: Yes

PLOS authors have the option to publish the peer review history of their article (what does this mean?). If published, this will include your full peer review and any attached files.

Reviewer #1: **Yes: **Martin Schaefer

Reviewer #2: No

---

## [Editor Report · Acceptance letter]

26 Jan 2023

PCOMPBIOL-D-22-00882R1 

Tissue specific LRRK2 interactomes reveal a distinct striatal functional unit

Dear Dr manzoni,

I am pleased to inform you that your manuscript has been formally accepted for publication in PLOS Computational Biology. Your manuscript is now with our production department and you will be notified of the publication date in due course.

With kind regards,

Bernadett Koltai
